# “I Would Never Come Here Because I’ve Got My Own Garden”: Older Adults’ Perceptions of Small Urban Green Spaces

**DOI:** 10.3390/ijerph16111994

**Published:** 2019-06-05

**Authors:** Vanessa G. Macintyre, Sarah Cotterill, Jamie Anderson, Chris Phillipson, Jack S. Benton, David P. French

**Affiliations:** 1Manchester Centre for Health Psychology, School of Health Sciences, University of Manchester, Manchester M13 9PL, UK; jack.benton@manchester.ac.uk; 2Centre for Biostatistics, School of Health Sciences, University of Manchester, Manchester M13 9PL, UK; sarah.cotterill@manchester.ac.uk; 3Manchester Urban Institute, School of Environment, Education and Development, University of Manchester, Manchester M13 9PL, UK; jamie.anderson@manchester.ac.uk; 4Manchester Institute for Collaborative Research on Ageing, School of Social Sciences, University of Manchester, Manchester M13 9PL, UK; christopher.phillipson@manchester.ac.uk

**Keywords:** urban greening, well-being, physical activity, qualitative, older adults, physical environment

## Abstract

Green spaces are known to improve health and wellbeing via several mechanisms, such as by reducing stress and facilitating physical activity. However, little is known about the impact of the smaller green spaces typically found in urban environments on wellbeing, especially for older adults. This study investigated experiences in adults (5 males and 10 females) aged 60 years and over of small urban green spaces in a large UK city. Fifteen older adults were interviewed using semi-structured walk-along interviews and photo elicitation methods in Old Moat, Greater Manchester. Twelve of the participants lived in Old Moat at the time of the study, and the remaining three participants previously lived in Old Moat and were frequent visitors. Transcribed interviews were analyzed using Thematic Analysis. Smaller urban green spaces were perceived differently to large green spaces, and participants were more likely to use larger green spaces such as parks. The smaller green spaces were perceived as belonging to other people, which discouraged the older adults from using them. The older adults also emphasized the importance of taking care of small urban green spaces and preventing them from becoming overgrown. Urban planners should consider these factors, since they indicate that the size and type of urban green spaces may influence whether they improve health and wellbeing. Further research should investigate in more detail which types of urban green space are most conducive to facilitating physical activity and improving wellbeing.

## 1. Introduction

The number of older adults in the population is increasing [1], and physical activity is lower in older adults than other age groups despite having many health benefits [2,3]. In addition, subjective wellbeing is an equally pertinent target for intervention since it is also low in older adults [4]. Subjective wellbeing is defined as an individual’s life satisfaction, happiness, and sense of purpose [5], and it can be improved by engaging in activities such as taking notice of one’s environment and interacting with others [6]. One promising intervention to increase both physical activity and wellbeing is to increase integration of green infrastructure into urban environments [7,8]. Green infrastructure (GI) is defined as an interconnected network of green space or natural features which supports ecological processes and provides benefits to human populations [9]. GI includes large parks, countryside, and urban woodlands, but can also consist of smaller areas of greenery such as street trees, private gardens, and grass verges [10,11]. The promotion of GI is promising and potentially more cost-effective than interventions targeted at individuals since it has a larger reach and thus can impact on a greater number of people [12].

Exposure to nature and green spaces has been found to increase health, happiness, and wellbeing through multiple pathways [13,14]. For example, whilst trees and greenery improve air quality by reducing air pollutants, green spaces facilitate physical activity, reduce stress, and provide opportunities for social interaction [14,15,16,17,18]. There is evidence that the quantity of public parks in cities predicts subjective wellbeing of residents [19]. Although quantitative evidence in older adults is lacking, in several qualitative studies, older adults have described increased feelings of wellbeing while spending time in green spaces and walking past street greenery [20,21]. In addition, older adults are more likely to walk on streets which are aesthetically pleasing [22], and greenery such as flowers and trees play an important role in improving the aesthetics of the environment [23]. Furthermore, since some older people are less able to travel to green spaces if they experience physical health problems, it is important for their wellbeing that their immediate surroundings provide opportunities to connect with nature [24,25]. In line with this, the presence of greenery in older adults’ private gardens is a significant predictor of wellbeing in this population [26]. Therefore, greater integration of urban green spaces and street greenery in cities may have the potential to increase physical activity and wellbeing in older adults. 

However, evidence of the influence of urban green space on older adults’ physical activity is scarce, and it is likely to differ from the findings on adult populations due to factors such as age and mobility [8,27]. In addition, a more in-depth understanding of how urban green spaces affect older adults’ health and wellbeing is necessary [28]. Qualitative research provides valuable insight into participants’ experiences, which can explain in greater depth how aspects of the environment impact on older adults’ lives [29]. However, none of the existing qualitative studies have focused on smaller green spaces in urban environments or examined in any depth how the aesthetics of the environment affect older adults’ experiences. In addition, most research has been typically conducted in the United States (US) and Australia, with few studies conducted in the United Kingdom (UK) [30,31]. The environment in the UK often has a higher urban density, as is typical in Europe, and the climate is often wetter and milder than countries such as US and Australia [8,32,33]. Therefore, existing research may not provide an accurate picture of the impact of green spaces on health and wellbeing in European countries with milder climates. Moreover, no qualitative studies to date have investigated older adults’ experiences of small-scale changes to urban green spaces in a built-up area or greater integration of urban green spaces in cities. Improved knowledge of the impact of urban green spaces could inform future changes to the environment that are aimed at improving the quality of life of older adults.

Experiences of change in urban green space have received little attention in qualitative studies. Some previous studies have conducted “walk-along” (or “go-along”) interviews [34], which is an approach that produces an in-depth understanding of how participants perceive their environment, since it involves talking about their experience of the environment as they walk through it [35,36]. This approach has been used to examine how street infrastructure, such as the condition of pavements and streets, can facilitate or inhibit older adults’ walking behavior [37,38]. However, no previous studies using walk-along interviews have specifically focused on the influence of small urban green spaces on older adults’ physical activity and wellbeing, or on older adults’ subjective experiences of these spaces.

The purpose of this study was to investigate older adults’ subjective experiences of small urban green spaces in a large UK city, using a walk-along interview approach. Where it was not possible to conduct walk-along interviews, photo elicitation methodology was used, which is an approach that produces a different kind of data to verbal interviews since it involves processing visual information [39]. Walk-along interviews and photo elicitation methodologies are comparable in terms of their objectives, since they both enable participants to experience the environment to some extent as they are talking about it. In walk-along interviews, this involved participants answering interview questions as they walked through the green spaces, whereas photo elicitation involved participants answering questions as they viewed pictures of the green spaces. The present study had several specific objectives. Firstly, we were primarily interested in investigating older adults’ experiences of small urban green spaces. Secondly, we investigated whether visiting these spaces affected older adults’ wellbeing. The third research question was how the presence of small urban green spaces influenced older adults’ physical activity.

## 2. Materials and Methods 

### 2.1. Study Design

The study had a cross-sectional design, in which semi-structured qualitative interviews were conducted using walk-along and photo elicitation methods, both of which enabled participants to view green spaces as they answered the interview questions. Our epistemological approach was grounded within the theoretical position of realism, a stance which assumes that language directly reflects experience [29]. This approach was considered to be most suitable since our focus was on participants’ individual experiences and our assumptions were that participants are able to communicate their realities effectively.

### 2.2. Location of Urban Green Space Changes

The study was set in Old Moat, part of the Withington area in Manchester (GM), a large city in England. Manchester is undergoing major economic development and increasing urbanization and is therefore a priority for urban street greening [40]. Old Moat was considered to be particularly suitable for the investigation of the impact of urban green spaces on older adults’ wellbeing, since Old Moat is a deprived area with a large population of older adults [41,42]. In recent years, 20% of the population of Old Moat were experiencing economic deprivation [43], and 11% of residents were aged 60 or over [44]. Changes to small urban green spaces in publicly accessible areas of Old Moat have recently been implemented by Southway Housing Trust, a housing association in GM. The changes included tree and flower planting and artificial tree decorations, with the aim of improving wellbeing in the community and increasing the use of sites which have undergone changes by residents of Old Moat [7,8]. The urban green spaces are small, ranging from approximately 0.05 to 0.50 acres.

### 2.3. Participants

A sample of 15 older adults was recruited from community groups within Old Moat and Withington. In order to recruit participants, the researcher attended the community groups and gave a short presentation providing information about the study, and spoke individually to members of the groups to find out if they were interested in participating. In addition, an advertisement of the study was published in the Withington Civic Society newsletter, inviting residents of Old Moat who were eligible to participate and interested in the study to contact the researchers. Any adults aged 60 or over in 2018 were considered eligible to participate if they lived or spent a large amount of time (i.e., a minimum of one or more hours every two weeks) in Old Moat when the study took place. Participants were excluded from the study if they had a diagnosis of dementia, since this could affect their ability to participate in the interviews. Participants who were able to complete walk-along interviews were prioritized for recruitment, although participants who preferred to participate in a sitting down photo elicitation interview were also recruited. Ten of the participants completed walk-along interviews and five participants completed photo elicitation interviews.

### 2.4. Procedure

Participants were approached at community groups in Withington, such as the Naturally Occurring Retirement Community (NORC) group, and Withington Assist, a group providing support and volunteering opportunities for older people. A “snowball sampling” technique was also utilized, which involves participants volunteering to encourage their friends and family to participate. Participants who were interested in participating were provided with a Participant Information Sheet (PIS) and were contacted by the researcher to arrange an interview. 

Participants who were happy to walk around Old Moat were interviewed using walk-along interviews. This consisted of walking to areas of Old Moat where changes to green spaces have taken place within the past 10 years. For each interview, the researcher met the participant in a location of their choice. Following consent, the researcher and participant agreed on a walking route, and the interview began from that location. The walk-along interviews involved asking questions as the researcher and participant walked along the streets of Old Moat and through the urban green spaces. On many occasions, it was necessary to stop and spend time looking at the sites, in order to ask more detailed questions about participants’ experiences of the sites before continuing the walk. Participants who did not walk completed a face-to-face photo elicitation interview while sitting in their homes. At certain points in the interview, these participants were presented with photos of urban green spaces which had undergone changes, and their views on the changes were explored. For all participants, the interview was audio recorded using an encrypted Olympus audio recording device and transcribed verbatim. Interviews lasted between 30 and 100 min.

### 2.5. Walk-Along Interview Routes

Urban green spaces and streets in Old Moat where changes had taken place over the past 10 years were investigated in this study and selected for inclusion in the walk-along interviews. All of the selected intervention sites were included in a plan of two potential walk-along interview routes, which could be adapted depending on individual circumstances but formed the basis for all routes taken during the walk-along interviews. The two potential walk-along interview routes in the plan were risk assessed, and the length of time taken to walk on the routes at a comfortable pace was measured to ensure that it did not take longer than one hour. The walking route varied depending on participants’ ability to walk and the location where the researcher and participant met. However, all walk-along routes included as many sites as possible which had been identified as having undergone changes in the past 10 years. Maps of the two potential walk-along interview routes are shown in Figure 1.

### 2.6. Photo Elicitation Methodology

Fifteen A4 color photos were taken of the selected intervention sites in Old Moat, all of which were presented to participants during sitting down interviews. Examples of the photos are shown in Figure 2. In addition, maps of the two potential walk-along routes were included. Participants were shown the location of the sites on either of the maps if they did not recognize any of the sites or were unsure of their location.

### 2.7. Interview Topic Guide

A semi-structured interview topic guide was developed which consisted of open-ended questions exploring participants’ experiences both of the changes to urban green spaces in Old Moat, and that of other areas. Some of the questions covered in the topic guide related to general issues about participants’ use of the green spaces in Old Moat, whilst others were more specific, concerning their views on certain aspects of the intervention sites. For example, at the beginning of the interview, the researcher asked questions such as “How do you usually get around your local area?” and “Have there been many changes to the area in the past 10 years?”. In contrast, whilst visiting specific sites, the researcher asked questions such as “What is it about this site that makes it attractive to use?”, “Is there anything you dislike about this site?”, and “Are there any ways in which this space could be improved?”. In addition, in order to compare participants’ experiences of the urban green spaces in Old Moat with their experiences of green spaces in other areas, the researcher asked, “What did you like about this green space compared to other green spaces?”. The researcher also asked participants the question “There have been a number of recent changes to Old Moat—which of these changes have you noticed?” before visiting any of the sites or showing participants photos of sites. This gave participants an opportunity to recall any experiences which came to mind before being prompted by seeing the sites. It also ensured that the data provided a true reflection of which green space changes participants found most noticeable, and that the data were not distorted by the researcher’s preconceptions.

### 2.8. Analysis

The data were analyzed electronically using QSR NVivo (Version 12, QSR International Pty Ltd, Doncaster, VIC, Australia) for Thematic Analysis, a flexible approach which identifies frequently occurring themes and enables rich interpretations of the data to be made [29]. This type of analysis was most appropriate since the research question focused on experiences of urban green spaces across a larger sample, as opposed to a more in-depth analysis of a few individuals’ experiences. Coding of the data was completed by the first and last authors of the project, and discussed with other members of the research team with expertise in qualitative research and urban greening. Preliminary codes were identified using a primarily inductive approach, which involves linking the themes directly to the data rather than making interpretations of deeper meanings [45]. This method was preferable since it is compatible with a realist perspective [29], and we did not have any preconceptions about the mechanism by which changes to urban green spaces affect older adults’ experiences. However, our approach was also partially deductive since we specifically focused on data which appeared to address our research question of how participants’ experiences are affected by the green spaces. The themes identified included both semantic and latent dimensions, since we were primarily interested in experiences explicitly described by participants, but some interpretations of their speech were necessary. The identification of latent themes enabled the research team to make sense of recurring themes which were interesting and relevant to the research question, but which were not explained explicitly by participants.

### 2.9. Ethical Approval

Ethical approval was granted for this study by the University of Manchester Proportionate Research Ethics Committee (UREC reference: 2018-4390-6568). Data were anonymized at the point of transcription. 

## 3. Results

A total of 15 participants were interviewed, consisting of 5 males and 10 females, as shown in Table 1. Twelve of the participants lived in Old Moat at the time of the study, and the participants who no longer lived there spent a great deal of time in Old Moat visiting friends or relatives and attending community events. As a result, all participants had recent experiences of being in Old Moat and were familiar with the area.

Three themes were identified. Overall, they indicated that older adults viewed the small urban green spaces as areas which are intended for use by people other than themselves. Participants were more likely to use larger, more public, green spaces where there are activities to engage in, such as visiting boating lakes or cafés, or looking at tropical flowers and wildlife. However, the older adults also appreciated the presence of urban green spaces in Old Moat, and felt that they improved the area provided that they were properly maintained. A summary of the themes and subthemes is shown in Table 2.

### 3.1. Theme 1: Small Urban Green Spaces Are Not for Them

Participants frequently referred to the urban green spaces as areas which are for people to use other than themselves, such as residents who lived adjacent to the green spaces. This often consisted of speculating on how other people would benefit from using the green spaces. For example, participants commented on whether residents of houses next to urban green areas would use them and how they would be affected by them:

“This is their little island, isn’t it?... For these houses, which looks nice doesn’t it? If you lived here, it looks nice. It makes you want to you live here.” (Margaret)

“Well peoples opposite, you know, if you live there and you’ve got nice green outside and it’s maintained, it’s nice isn’t it? You’re looking out the window to a bit of green where if you’re overlooking something like a brick house, it’s a bit more… it doesn’t want to make you go out, you just feel like you want to stay in all the time… You think oh that’s nice, you know, if you was buying a house like you see all these nice houses that you want to buy. You go round and think oh they’re nice houses but they’re all facing each other. But when you see something like this, suddenly it makes you want to go live there doesn’t it? Because you think you’ve got your own park, got my own space, yeah.” (David)

Some participants even expressed the view that the purpose of the green spaces was to improve social cohesion by providing a place for residents living next to green spaces to sit together and socialise:

“I would design chairs, comfortable seats that, well maybe comfortable benches facing each other, so that a whole family could either have a picnic or just talk to each other under a shady tree and, not just families but elderly people, who may be a widow or a widower, want to sit under a tree and sit face to face with a neighbor, who might look out of the window and say ‘oh Nelly’s there, I haven’t seen Nelly for ages cause I never see anybody, I’m in the house all the time, and they’ve got some new chairs so I’ll go and sit facing Nelly’, and then they’ll probably do that every day for life then, they’ll become good friends and that’s what social housing is about, social cohesion.” (Alan)

Others discussed how children could use the areas, and often appeared to assume that the green spaces were intended for children to use:

“You’ve got your big boulders which the children could probably sit on. A bit low for the likes of me, you know.” (Agatha)

“Oh lovely. Yes and… I’m sure the children used to play on this green. Course there’s never children playing on it now but I suppose there are at times… [What do you think about that, children playing?] Yeah, lovely, yes and cause I do like, and there’s another green around here… There were lots of greens around.” (Muriel)

Participants also revealed that they were unaffected by changes to the urban green spaces since they did not live next to them:

“On this particular one, because it’s not very big, is it, so they’d soon wreck it. I mean kids do go on it but they run round, they don’t... There’s nothing to play… It’s just plain, isn’t it?… It wouldn’t bother me cause I live here [a few houses away, green space still visible from her house] so I’m not involved up there… It doesn’t bother me cause it, I don’t go up there, I don’t.” (Ethel)

Furthermore, some participants admitted that they would be discouraged from using the urban green spaces in their area because they felt that these were intended for residents who lived close by. They preferred to visit larger spaces which were understood to be intended for wider public use:

“… And maybe the people who live around those places are the ones who feel like it’s their own, I don’t know, yeah, it’s not somewhere I could come from my house and just come round those places, you know, someone would be phoning the police saying oh, there’s somebody dodgy [laughs], you know… It’s sort of… yeah, you feel those green spaces, they seem like they belong to that group of houses… There is one in Didsbury… it’s bigger, isn’t it, and it’s more public, you know, people… anybody can go there without feeling you’re intruding on other people.” (Doris)

“Well as a single person, for me to get the bus there, I could go and sit on that lump of a tree there. You know, I don’t know anybody there sort of thing and, ‘who’s that old fella sat there like’, you know, I’d feel a bit conscious… no I don’t, I don’t think I would wander down there.” (Roderick)

Lastly, green spaces were often referred to as somewhere to sit and relax rather than places to walk through. Green spaces which did not have benches, such as some of the smaller urban green spaces in Old Moat, were viewed as less useful. For some, this was due to problems with mobility, whereas for others, it was because there was nowhere to sit and relax:

“I’ve got me own garden so I don’t need to. I do know there are quite a few of them that go and sit outside the café, but I don’t need to do that because I’ve got a garden… It’s nice for anyone that wants to sit and, you know, if it’s a nice day you can sit and look round and that, see what’s going on. See what’s happening. It’s better than sitting in the house.” (Agnes)

“… They could make as many benches and seating as they want and there’ll never be too many seats cause people like me can only go so far and want to sit down. [So you wouldn’t come down this way do you mean?] Not if I didn’t have my trolley because I don’t know how far it is.” (Emily)

“… That’s a wasted opportunity, they’ve got mature trees that can provide shelter from the sun and occasional rain showers and not one of them has a bench under it… You don’t need those new trees, you need benches. Some in the sun and some in the shade. Some people would want to sunbathe on a bench and others would want to, like me, I want to sit in the shade…” (Alan)

### 3.2. Theme 2: Differences in How Larger and Smaller Green Spaces Are Perceived

The majority of older adults perceived the small urban green spaces differently to how they perceived larger green spaces, although this view was not expressed by all participants. For example, participants preferred larger spaces and felt that the benefits of smaller urban green spaces were limited due to their size:

“We’ve got these areas at the end of the roads on the estate, which are very very pleasant, there’s no doubt about it, got nice benches… but yesterday I was at Whitworth Art gallery... I was at a meeting, and I looked out of the window and there’s a huge green, not Platt Fields although that’s there, but belonging to Whitworth Art Gallery, there’s a huge field there, and I looked out and there’s all the families with the kids having a picnic, you know… And I think that’s nice… In the actual Old Moat ward, that’s how they refer to these things, other than the things I’ve mentioned, which is the Old Moat Park and that, no there’s not, you know, a lot of green spaces. Where there’s a bit of land, Southway, credit to them, have tried to make it inviting for people and that, you know.” (Roderick)

“They’ve landscaped it and put all flowers in it and everything. Yeah. [Is there anything else that they’ve kind of done that you’ve particularly liked?] No, there’s not much they can do with it is there? When it’s just a grass verge, there’s not much they can do with it, only make it pleasant and tidy, keep it tidy and… there’s nothing really else they can do with them.” (Beryll)

Participants described the smaller green spaces as having a positive impact on the area, but would not choose to visit or walk past them. When they compared these with larger spaces, they talked about there being more to see and do in larger parks, and that this was the reason for their preference:

“Sometimes they’ve got things in it [Chorlton Water Park] rather than just look at. There’s something to do, either potting or animals or flowers to look at, whereas these just got a green space, don’t you?... You can walk the dogs, there’s a swing area for the children. There’s toilets there if you need a toilet, and there’s a picnic area if you want to take food or you want to have food, and there’s swans and ducks and things like that, and wildlife to look at. So it’s not just a green space, is it? ... To me, these green spaces [referring to smaller urban green spaces] only look nice. There’s no... Like say this here, but what’s it here for? It just looks better as you’re walking through the estate but it doesn’t do anything… It looks tidy, and you do think you’re walking through a green space, but that’s all you’re doing, you’re walking past it. It’s just there, rather than a rubbish tip… I do change my route, but not for scenery or anything. I don’t think I would for the scenery. Only for a change for me, thinking I’ll go up that way this time, rather than going up that way.” (Ethel)

In addition, even though the spaces in Old Moat were primarily viewed as places to sit rather than places for walking, some participants described going for walks in larger green and blue spaces in other areas in and around Manchester:

“I don’t walk too much around these places. I… when I like walking, like I go, I go walking different places, but if I’m going for a walk I get a bus up to Didsbury and I go for… I go on the, for a walk on the River Mersey.” (William)

“I would never come here because I’ve got my own garden, and I’ve got a car that will take me to Dunham Massey. I’m a national trust member so Dunham Massey is my private garden, you know the deer park. I can go there [laughs] it’s a lot bigger… It’s free for members and I’m a member, and I love taking friends and visitors to admire it.” (Alan)

### 3.3. Theme 3: Taking Care of the Area and the Role of Green Spaces

Despite expressing views that green spaces were intended for use by other people and that smaller urban green spaces had limited use, the older adults also described the importance of having greenery in a built-up environment, and how this influenced their perception of the area:

“What you have down there on the lane is some recesses, they’re like circular grass areas with the houses set back from… and I love that because when you’re walking down the main road you’ve not just got houses, houses, houses, you’ve always got a little break every so often with somewhere pleasant. I have often wondered whoever owns the land why they’ve not built on it, but then I’ve been delighted that they haven’t and thought, well they’re not greedy, cause most individuals would put half a dozen houses there… So in a way it’s a luxury to have grass and trees and that set back area in front of your house and if they can still afford to do it well that… that’s special to Withington...” (Emily)

However, whilst participants appreciated having trees in a built-up area, they discussed the disadvantages of planting too many trees:

“The mature trees really do bring a special energy to this place, and sitting under a mature tree is a wonderful feeling but there’s nowhere to sit except on the ground… I mean just look, you can sit here and look through the dappled sunlight of the leaves. It’s amazing but if you plant other trees around it, you won’t see the sun. It would block out the sun, just through these mature trees. So planting trees too close together is a big mistake… new trees are fine if there’s room for them but I don’t think new trees need lots of light... So these trees are unhappy because they’re in shade, and when they do get big enough, they’re going to be interfering with the mature trees.” (Alan)

“If you’ve got a house and you ring up the insurance, ‘Any trees?’, ‘What do you mean trees?’, ‘Any trees within a certain distance?’ There’s a certain distance I use, I don’t know how many yards it is but, er, I think if they go on the websites and find out the tree has torn your house, they get a bit worried and it’s too close because they think, well hang about, you know what insurance companies are like don’t you? Let’s rip you off, oh we can rip them off for this now because the tree’s too close to the house so they put your insurance up… Trees, good and bad ideas, you know, they can cause a lot of problems with the roots and that, as well as blocking lights out. And branches falling off in the wind and smashing people’s cars which are parked on the road, you know, all kinds of stuff like that. So you’d have to kind of make sure the trees are a certain kind that don’t have them effects, you know, they don’t grow too high in that particular area.” (David)

In addition, the need to look after and maintain green spaces was frequently emphasized:

“It looks nicer it’s… yeah it looks nice, down there. There’s no bin here [laughs] no rubbish bin, that would be nice, wouldn’t it, you know if people throw things around… I don’t know if there ever was but I’ve just noticed that can there, but yes I do like it, I suppose it’s neatness and it looks nicer.” (Muriel)

Many participants described the importance of cutting the grass to ensure that the area did not become overgrown:

“Well it would be overgrown, it would smother itself more or less you might say… you see the grass gets too high and you’re not getting the sun.” (Roderick)

“It looks as though it’s been given a bit of attention doesn’t it, and it’s not been, before it was neglected, it was just allowed to just grow wild. But now it’s been brought, you know, into the community to look nice, yeah and it does.” (Colin)

Some participants described how the general impression of the area is affected by whether or not public urban green spaces are taken care of:

“Round the corner where the greens are, they had been brought up tremendously and they’ve all been re-turfed and trees planted and I think there’s a picnic table round there. A big difference to the green spaces and it makes the place look better when you’re driving in because before there was cars parked on there or the kids were playing football on it and things like that.” (Jean)

In addition, the older adults felt that it is equally important to ensure that greenery in private properties is properly maintained, since it also affects public perceptions of the area:

“It just gives a better impression to the area, you know. Behind you they’ve recently cut the grass on that property there. You could almost put some horses or some sheep in because the… somebody had just neglected it for so long and it just didn’t look nice. It didn’t look nice. It gives a false impression to the area, you know. If it’s looked after, people think, people that come past could admire it, you know, in that sense. No, I think it looks neater. It’s better. It just gives a better impression to the area if it’s being looked after. You know, everybody’s looking after bits and pieces, you know, for… people like to live in a nice area.” (Agatha)

Participants also felt that if the area was maintained, anti-social behavior such as dropping litter was less likely to occur:

“I think if it’s wild, people start dumping stuff because they think, oh it’s a tip let’s go dump something. Oh we can hide that in that grass over there, just chuck the bike frame in there or the old, you know, it starts to become a dump then, wouldn’t it? But if it’s neat, it stops people doing that because it’s neat, you know, you can’t hide anything and it stands out doesn’t it?” (David)

“It doesn’t look neglected, it looks owned doesn’t it?... And I think that sort of thing is important, because I think it makes people look after it more, if this was left to grow a bit wild, people would start perhaps dumping their rubbish here... [How would you feel if it was neglected, how would you feel about that?] That would be awful, because that just brings the whole tone of an area down, and also it’d bring the house prices down, it would encourage things like fly tipping and vandalism. I think it’s all part and parcel of this, if areas are owned and looked after, and somebody takes a bit of pride in them, it discourages antisocial behavior. I think, personally, I think the more you put in, and it’s little things, like putting in a bit of artwork, and keeping the grass mowed, it’s just little things like that encourage not antisocial behavior.” (Margaret)

In addition to looking after green spaces, the importance of taking care of the entire area was emphasized, including maintaining the roads and providing enough facilities such as supermarkets and shops. This was described by many participants as just as important as taking care of green spaces:

“It [Withington Village] should have a bit of an identity instead of just charity shops and, I mean it’s trying, it’s got a café, it’s got a nice cake shop and it’s got a Co-op, but I think the other things in between are a bit erm, I don’t know… I think the only thing they can do down there is start having a bit of a supermarket, an Aldi or something... They just seem to have forgotten the high street, you know, the village, they seem to have forgotten.” (Christine)

## 4. Discussion

The present study explored older adults’ experiences of small urban green spaces in a large UK city using walk-along interviews and photo elicitation methodology. Three themes were identified. Firstly, older adults viewed the small urban green spaces in their area as belonging to people other than themselves, typically those who lived close by. Secondly, they were less likely to make use of small urban green spaces in their area, and were more likely to use larger parks. This was partly due to their perception that these spaces were more public, and partly because of their size and attractions, such as lakes, wildlife, and tropical flowers. Lastly, participants emphasized the importance of maintaining small urban green spaces, such as by cutting the grass to prevent it from becoming overgrown. They felt that if these spaces were maintained properly, residents were more likely to take care of the area and less likely to drop litter or neglect it in other ways. All three themes link to each of the study aims. The first theme suggests that older adults’ experiences of the small green spaces are affected by their perception that the spaces are intended for use by other people, and the second theme suggests that they may be less likely to use smaller green spaces than larger green spaces. As a result, it appears as though the presence of smaller green spaces is unlikely to increase older adults’ subjective wellbeing or physical activity. The third theme suggests that maintenance of the area affects older adults’ experiences of the green spaces and their subjective wellbeing, which in turn may influence whether older adults choose to walk near the green spaces.

The study had three main strengths. Foremost, the use of walk-along interviews enabled most participants to experience the urban green spaces as they responded to interview questions. In addition, the use of photo elicitation interviews enabled older adults who were unable to participate in walk-along interviews to view the green spaces while answering interview questions. As a result, participants noticed and discussed aspects of the green spaces which may not have occurred to them during a verbal interview that did not use either of these methodologies. Therefore, these novel approaches produced a more in-depth account of participants’ experiences of the urban green spaces. Lastly, the study took place in a UK city, which addresses gaps in the literature on the impact of urban green spaces in European cities with milder climates.

The study had some limitations. Firstly, even though the use of photo elicitation methodology has significant advantages, it would have been preferable if all participants had been able to participate in walk-along interviews. Since this was not possible, some of the participants were not able to experience all sensory aspects of the urban green spaces as they were talking about them, such as sounds of birds or smells of certain flowers. However, this is unlikely to have affected the findings since the visual aspects of the urban green spaces were most relevant to the research questions, and they were captured in the photos used for the study. An additional limitation is that three of the participants did not live in Old Moat at the time of the interview. However, they had sufficient experience of spending time in the area to be able to express their views on the urban green spaces there. All three participants regularly attended the Naturally Occurring Retirement Community (NORC) group and other community groups held in Old Moat at the time of the study. The NORC group meet once every fortnight for two hours, but often organize additional events which also take place in Old Moat. Lastly, Old Moat is an area where a number of changes have taken place over the past 10 years, such as urban green space interventions and the addition of benches on streets [41]. Therefore, it is possible that these previous interventions have influenced participants’ experiences of recent changes to the urban green spaces in Old Moat, particularly in terms of whether they noticed or appreciated the changes. In an area where no previous interventions have taken place and where urban green spaces have only been recently introduced, residents’ experiences of these urban green spaces may be different. This is a limitation which can be addressed by further research on people’s experiences of urban green space integration. However, given that these previous interventions have taken place in Old Moat, it may have been expected that participants would be aware that the small urban green spaces in Old Moat are intended for public use. It may be that participants viewed particular spaces as belonging to certain people due to the “garden suburb” layout of Old Moat, which has large amounts of private green space, and its stable long-standing population. This is a topic that future research could investigate.

Previous literature on the impact of urban green spaces on wellbeing and physical activity has revealed that nature and green spaces facilitate physical activity in adults, and promote happiness and wellbeing in both adults and older adults [13,14]. Older adults have described feelings of increased wellbeing in urban green spaces such as large parks, allotment gardens, and while walking past greenery such as trees and flowers [20,21,46]. The present study adds to this literature by considering the specific impact of smaller urban green spaces on physical activity and wellbeing. 

The present study revealed factors which influence older adults’ use of urban green spaces, and identified types of urban green space which may have fewer benefits. Our finding that older adults perceived smaller urban green spaces as being intended for use by people other than themselves relates to previous literature on place identity. Place identity is the extent to which individuals identify themselves as belonging to a particular place [47]. This is a concept which has been investigated in relation to urban green spaces in quantitative studies [48,49]. However, older adults’ experiences of place identity in urban green spaces has not been explored qualitatively in previous literature. It may be that experiences of place identity are affected by both the sizes of urban green spaces and their geographical location. If an urban green space is immediately adjacent to an older adult’s house, they may perceive it as belonging to them, whereas if it is next to other houses, they may perceive it as belonging to the residents of these other houses.

Similarly, participants’ preferences of walking in larger parks as opposed to walking past smaller urban green spaces relates to literature on place dependence. Place dependence is defined as the extent to which a particular place enables people to achieve their goals [50]. It may be that older adults are more likely to seek out green spaces which have attractions such as lakes, wildlife, tropical flowers, and cafés, and that larger urban green spaces are more likely to have these features. In contrast, urban green spaces which are too small to contain these features may not be perceived as worth visiting since they do not enable individuals to achieve this goal. This is consistent with previous findings that older adults view physical activity as a consequence of engaging in other activities [51]. It may be that for older adults, the main aim of visiting larger green spaces is to visit these attractions, and that any physical activity they engage in is as a consequence of that goal. Therefore, the presence of urban green spaces may not provide a sufficient motivation to walk or cycle in their area since they do not contain attractions such as lakes, tropical flowers, or cafés. 

The older adults’ preference for well-maintained urban green spaces is consistent with previous research indicating that residents perceive overgrown grass in their area as having a negative impact, such as decreasing their sense of safety [52]. This study expands on that literature with the finding that older adults felt that people are more likely engage in antisocial behavior such as dropping litter in urban green spaces if they become overgrown.

These findings have several important implications for practice. First of all, they indicate that while the integration of urban green infrastructure into residential urban areas is often beneficial, it also has some limitations. Specifically, urban green spaces may need to be sufficiently large to increase the likelihood that older adults will use them to engage in physical activity. In addition, urban green spaces in residential areas which are not perceived to be for public use (e.g., due to lack of seating areas) are less likely to facilitate physical activity. The need for larger urban green spaces to facilitate physical activity is likely to pose challenges for urban planners, due to UK and other European cities becoming increasingly crowded and densely populated. There is likely to be competing interest for space for the development of new houses and flats, which limits the practicality of developing many larger urban green spaces in cities. One potential solution to address this problem could be to improve the connectivity between multiple smaller urban green spaces, as this may encourage people to use these spaces. For example, the presence of footpaths and signs indicating that particular urban green spaces are linked together may improve their connectivity. However, it may equally be the case that smaller urban green spaces do not facilitate physical activity under any circumstances, and that urban planners need to think carefully about alternative uses for these spaces. It may be more beneficial to use small urban green spaces for community led interventions such as mobile libraries and community cafés, which have the potential to impact on the wellbeing of nearby residents. However, despite these limitations, smaller urban green spaces do contribute to older adults’ wellbeing by having a positive impact on their perceptions of the area, provided that the greenery appears to be taken care of. Therefore, any small urban green spaces which are integrated into cities should be well-maintained, such as cutting the grass regularly, otherwise they may negatively affect residents’ perceptions of the area.

Further research should investigate in greater detail to what extent the geographical location and specific features of urban green spaces deter older adults from using the spaces. For example, whether or not the urban green spaces are in a residential area may have an impact on older adults’ perceptions of the space belonging to other people. Secondly, whether or not the green spaces are surrounded by fences may influence older adults’ perceptions of whether the space is for public use. Furthermore, future studies should establish how large an urban green space would need to be, and what attractions it would need to contain, to be considered to be worth visiting by older adults. Moreover, further research should empirically test the hypothesis that older adults’ experiences of smaller green spaces differ from their experiences of larger green spaces using quantitative methods. Studies should also investigate whether improving the connectivity between multiple small urban green spaces has an impact on physical activity in older adults. In addition, future research should investigate which specific features of green spaces are most likely to facilitate and improve behavioral indicators of wellbeing, such as interacting with others and taking notice of one’s environment. The use of focus groups may prompt discussions between participants on their use of green spaces, which could produce new insights into the reasons for differences in their experiences of larger and smaller green spaces. Additionally, this could prompt discussions on the history of the area, which may reveal how aspects of the socio-cultural context affect participants’ use of the green spaces. However, focus groups would have been less appropriate for our study since we were more interested in examining individual participants’ experiences of recent changes to specific green spaces. Moreover, future studies should investigate whether similar findings are produced in other UK cities and in other countries. This should include research in areas with different socio-cultural contexts, such as lower urban densities, more transient populations, other housing types, and warmer climates, since findings may be affected by these factors. For example, the impact of environmental attributes such as the quality of pavements on physical activity in older adults varies between high and low-middle income countries [53]. Therefore, older adults’ use of small green spaces in different socio-cultural contexts may be influenced by multiple factors including the stability of the population, urban density, and whether the area contains private or social housing. Further research should also include more deprived areas which have not undergone any previous interventions, in order to investigate whether older adults in these areas have similar views on urban green spaces. Lastly, these findings should be incorporated into current approaches to assessing the value of urban green spaces. Recent approaches have mapped the density and spatial coverage of green space, including its features such as trees and vegetation [54]. However, once further research has been conducted to support our findings, features of urban green spaces which affect individuals’ experiences of place identity and place dependence should be included in these assessments. This would enable urban planners to make more informed decisions on the types of urban green spaces which should be integrated into cities, and to evaluate their potential impact.

## 5. Conclusions

In the present study, adults viewed the small urban green spaces as belonging to people other than themselves, and preferred to visit larger green spaces. In addition, they had a strong preference for urban green spaces which are maintained regularly, and felt that overgrown green spaces have a negative impact on the area. These findings have implications for urban planning, and suggest that urban green spaces may only increase physical activity and wellbeing in older adults under certain circumstances.

## Figures and Tables

**Figure 1 ijerph-16-01994-f001:**
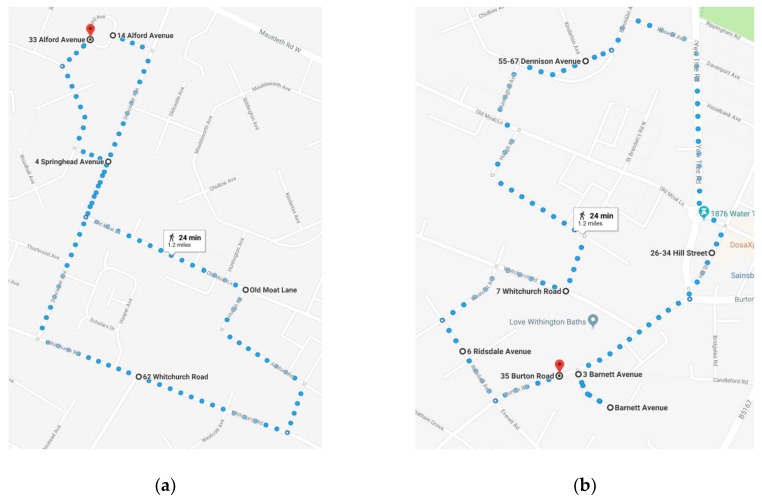
Maps of two potential walk-along interview routes. (**a**) A route which includes urban green spaces on Old Moat Lane, Alford Avenue, and Whitchurch Road; (**b**) A route which includes urban green spaces on Dennison Avenue, Old Moat Lane, Parbold Avenue, and Burton Road.

**Figure 2 ijerph-16-01994-f002:**
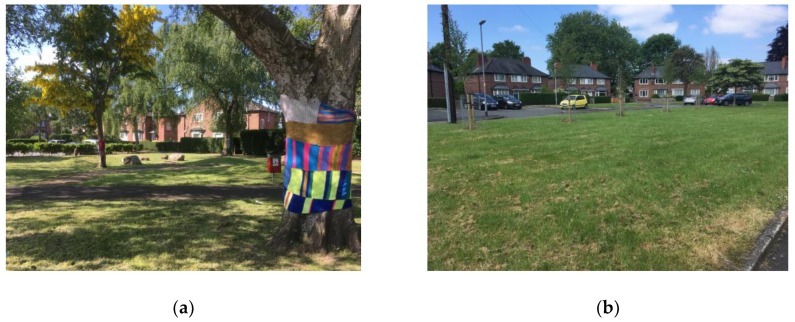
Photos used in the photo elicitation interviews. (**a**) The green space on Old Moat Lane; (**b**) The green space on Parbold Avenue. (Both photos were taken by Vanessa Macintyre).

**Table 1 ijerph-16-01994-t001:** Participant demographics.

Participant	Experience of Old Moat	Type of Interview Conducted
Christine	Used to live in Old Moat and still spends time there	Walk-along
Emily	Used to live in Old Moat and still spends time there	Walk-along
Alan	Lives in Old Moat	Walk-along
Muriel	Lives in Old Moat	Walk-along
Jean	Lives in Old Moat	Photo elicitation
Colin	Lives in Old Moat	Photo elicitation
Roderick	Lives in Old Moat	Photo elicitation
Margaret	Lives in Old Moat	Walk-along
Beryll	Lives in Old Moat	Photo elicitation
Agatha	Lives in Old Moat	Walk-along
William	Lives in Withington (an area adjacent to Old Moat) but spends a lot of time in Old Moat	Walk-along
Ethel	Lives in Old Moat	Walk-along
Agnes	Lives in Old Moat	Photo elicitation
David	Lives in Old Moat	Walk-along
Doris	Lives in Old Moat	Walk-along

**Table 2 ijerph-16-01994-t002:** Details of key themes.

Theme	Number of Participants the Theme Was Shared Across	Subthemes	Example Participant Quote	Number of Participants Who Produced Material Included within Each Subtheme
Theme 1: Small urban green spaces are not for them	15	Small green spaces are intended for people living adjacent to them	“This is their little island, isn’t it?”	9
		Small green spaces are intended for children	“You’ve got your big boulders which the children could probably sit on.”	11
		Green spaces which do not have benches are less useful	“... you need benches. Some in the sun and some in the shade.”	10
Theme 2: Differences in how larger and smaller green spaces are perceived	9	Larger green spaces have more attractions than small green spaces	“When it’s just a grass verge, there’s not much they can do with it.”	9
		Larger green spaces are better places to walk in	“... if I’m going for a walk I get a bus up to Didsbury... and I go... for a walk on the River Mersey...”	5
Theme 3: Taking care of the area and the role of green spaces	15	Importance of maintaining green spaces	“...keeping the grass mowed, it’s just little things like that encourage not antisocial behavior”	15
		The need for a balance between concrete and greenery in a built-up area	“... it’s a luxury to have grass and trees and that set back area in front of your house...”	8
		The importance of maintaining the entire area	“They just seem to have forgotten the high street...”	13

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
