# Peer review of "“I Would Never Come Here Because I’ve Got My Own Garden”: Older Adults’ Perceptions of Small Urban Green Spaces"

_ijerph, 2019, doi:10.3390/ijerph16111994_

Round 1

Reviewer 1 Report

This study aimed to investigate older adults’ experiences of small urban green spaces, and  whether visiting these spaces  affected their  wellbeing  and  amount of physical activity

The topic is interesting and I invite the authors to revise the paper.

Here my comment.

Abstract

Please indicate the N the main characteristics of the participants.

Line 42-43

Please think to change the sentence.

Lines 51 – 52

What  does 'improving air quality' mean

Lines 89-92

Please provide more information about the walking-alone and photo elicitation methodologies? Is it this two method comparable in term of results. The authors used these two methods interchangeably. I am wondering that this may be possible.

Line 93

I think that the aims of the study were three, not two.

Participant section

Please describe briefly the characteristics of the sample (e.g., mean age, number, etc. ). Moreover, I think that the author should better describe the recruiting process. How do the authors recruit 15 older adults?

Line 122

what does mean a large amount of time? How did the author measure “large amount of time”?

Discussion section: general comment

Rethink to restructure the discussion section as following:

-           Main findings

-           Discussion of the specific aim of the study :

o          older adults’ experiences of small urban green spaces,

o          older adults’ wellbeing.

o          older adults’ physical activity.

-           limitation

-           strength

Line 447

how did the authors measure the experience of older that did not live in Old Moa?

What about the difference with another socio-cultural context. Please deepen this aspect

Author Response

Point 1: This study aimed to investigate older adults’ experiences of small urban green spaces, and  whether visiting these spaces  affected their  wellbeing  and  amount of physical activity

The topic is interesting and I invite the authors to revise the paper.

Response 1: Thank you very much for your helpful comments and suggestions. We have followed many of these suggestions and feel that this has improved the quality of the paper.

Point 2: Abstract

Please indicate the N the main characteristics of the participants.

Response 2: The abstract has been changed to include the number of males and females in the sample and the number of participants who lived in Old Moat at the time of the study (lines 21-25):

This study investigated experiences in adults (5 males and 10 females) aged 60 years and over of small urban green spaces in a large UK city. Fifteen older adults were interviewed using semi-structured walk-along interviews and photo elicitation methods in Old Moat, Greater Manchester. Twelve of the participants lived in Old Moat at the time of the study, and the remaining three participants previously lived in Old Moat and were frequent visitors.

Point 3: Line 42-43

Please think to change the sentence.

Response 3: The sentence has been changed to improve its clarity (lines 43-44):

Subjective wellbeing is defined as an individual’s life satisfaction, happiness, and sense of purpose [5], and it can be improved by engaging in activities such as taking notice of one’s environment and interacting with others [6]

Point 4: Lines 51 – 52

What  does 'improving air quality' mean

Response 4: This sentence and the following sentence have been changed to clarify what was meant by this point (lines 53-55):

Exposure to nature and green spaces has been found to increase health, happiness, and wellbeing through multiple pathways [13,14]. For example, whilst trees and greenery improve air quality by reducing air pollutants, green spaces facilitate physical activity, reduce stress, and provide opportunities for social interaction [14-18].

Point 5: Lines 89-92

Please provide more information about the walking-alone and photo elicitation methodologies? Is it this two method comparable in term of results. The authors used these two methods interchangeably. I am wondering that this may be possible.

Response 5: More information has been provided on what is involved in the two methodologies and the reasons we believe that they produce comparable results (lines 98-102):

Walk-along interviews and photo elicitation methodologies are comparable in terms of their objectives, since they both enable participants to experience the environment to some extent as they are talking about it. In walk-along interviews, this involved participants answering interview questions as they walked through the green spaces, whereas photo elicitation involved participants answering questions as they viewed pictures of the green spaces.

Point 6: Line 93

I think that the aims of the study were three, not two.

Response 6: We have changed this section to include three aims: to investigate older adults’ experiences of small urban green spaces; to investigate whether visiting these spaces increase older adults’ wellbeing; and to investigate whether the presence of small urban green spaces increases physical activity in older adults (lines 102-106):

The present study had several specific objectives. Firstly, we were primarily interested in investigating older adults’ experiences of small urban green spaces. Secondly, we investigated whether visiting these spaces affected older adults’ wellbeing. The third research question was how the presence of small urban green spaces influenced older adults’ physical activity.

Point 7: Participant section

Please describe briefly the characteristics of the sample (e.g., mean age, number, etc. ). Moreover, I think that the author should better describe the recruiting process. How do the authors recruit 15 older adults?

Response 7: We have added more information about how participants were recruited. In addition, we have added information on the number of participants recruited and how many participants completed walk-along and photo elicitation interviews (lines 130-142):

A sample of 15 older adults was recruited from community groups within Old Moat and Withington. In order to recruit participants, the researcher attended the community groups and gave a short presentation providing information about the study, and spoke individually to members of the groups to find out if they were interested in participating. In addition, an advertisement of the study was published in the Withington Civic Society newsletter, inviting residents of Old Moat who were eligible to participate and interested in the study to contact the researchers. Any adults aged 60 or over in 2018 were considered eligible to participate if they lived or spent a large amount of time (i.e. a minimum of one or more hours every two weeks) in Old Moat when the study took place. Participants were excluded from the study if they had a diagnosis of dementia, since this could affect their ability to participate in the interviews. Participants who were able to complete walk-along interviews were prioritized for recruitment, although participants who preferred to participate in a sitting down photo elicitation interview were also recruited. Ten of the participants completed walk-along interviews and five participants completed photo elicitation interviews.

We do not report on participant age, as we did not collect this information. We decided against collecting data on participants’ ages during the interviews since Old Moat is a small area and most of the participants know each other. Therefore, we felt that reporting information about participants’ ages would be unethical since it could allow participants to be identified. As a result, it is not possible to calculate the mean age of all of the participants. However, six of the participants chose to mention their ages during the interviews. Of these six participants, their mean age was 76, and their ages ranged from 61 to 92 years old. All participants were aged 60 years or older.

Point 8: Line 122

what does mean a large amount of time? How did the author measure “large amount of time”?

Response 8: This sentence has been changed to clarify what we meant by a large amount of time (line 137):

Any adults aged 60 or over in 2018 were considered eligible to participate if they lived or spent a large amount of time (i.e. a minimum of one or more hours every two weeks) in Old Moat when the study took place.

Point 9: Discussion section: general comment

Rethink to restructure the discussion section as following:

-           Main findings

-           Discussion of the specific aim of the study :

o          older adults’ experiences of small urban green spaces,

o          older adults’ wellbeing.

o          older adults’ physical activity.

-           limitation

-           strength

Response 9: We are grateful for this suggestion and considered changing the structure of the discussion. However, we decided that because the three study themes cut across all of the study aims, it would not make sense to structure the discussion so that they are discussed separately.

Instead, we have clarified how each theme identified in the study addresses each study aim (lines 458-465):

All three themes link to each of the study aims. The first theme suggests that older adults’ experiences of the small green spaces are affected by their perception that the spaces are intended for use by other people, and the second theme suggests that they may be less likely to use smaller green spaces than larger green spaces. As a result, it appears as though the presence of smaller green spaces is unlikely to increase older adults’ subjective wellbeing or physical activity. The third theme suggests that maintenance of the area affects older adults’ experiences of the green spaces and their subjective wellbeing, which in turn may influence whether older adults choose to walk near the green spaces.

The structure we have used in the Discussion is in line with what has been previously recommended and widely adapted (Docherty & Smith, 1999).

Point 10: Line 447

how did the authors measure the experience of older that did not live in Old Moat?

Response 10: This has been clarified on lines 484-487:

All three participants regularly attended the Naturally Occurring Retirement Community (NORC) group and other community groups held in Old Moat at the time of the study. The NORC group meet once every fortnight for two hours, but often organize additional events which also take place in Old Moat.

Point 11: What about the difference with another socio-cultural context. Please deepen this aspect

Response 11: This potential difference has been discussed in greater depth in the limitations section on lines 494-499:

However, given that these previous interventions have taken place in Old Moat, it may have been expected that participants would be aware that the small urban green spaces in Old Moat are intended for public use. It may be that participants viewed particular spaces as belonging to certain people due to the “garden suburb” layout of Old Moat, which has large amounts of private green space, and its stable long-standing population. This is a topic that future research could investigate.

It has also been discussed in the further research section of the Discussion on lines 577-583:

This should include research in areas with different socio-cultural contexts, such as lower urban densities, more transient populations, other housing types, and warmer climates, since findings may be affected by these factors. For example, the impact of environmental attributes such as the quality of pavements on physical activity in older adults varies between high and low-middle income countries [54]. Therefore, older adults’ use of small green spaces in different socio-cultural contexts may be influenced by multiple factors including the stability of the population, urban density, and whether the area contains private or social housing.

Reviewer 2 Report

This is a very good paper and I would be happy to recommend publication with one or two small adjustments.

Firstly in the abstract lines 25-26 you should refer again to smaller urban green spaces being perceived as belonging to other people, rather than just green spaces.

Methodologically you state a clear case for the combination of tools employed but there is no evidence of further approaches which you rejected. This would display greater critical thinking methodologically and should be included.

Lines 179-184 The interviewer questions seem to presuppose a positive response? Were participants permitted to say that they did not find spaces attractive? 'What do you like about this green space compared to other green spaces'? Did you also ask - Is there anything you do not like about this space? Some clarification here is required.

How were the walk-round interviews recorded? Presumably using an audio recorder please clarify.

I understand that this was a qualitative study but I would be very interested to see a little quantification of the key themes and piling of the responses which fit within them. This could simply be a table which piles responses within sub-categories and then into the main themes. For instance within the theme of 'small urban spaces were not for them', what was the total amount of times that this was mentioned by participants? Clearly there were different reasons for this such as lack of amenity, lack of privacy etc. It would be interesting to see the frequency of these sub-themes within the main theme. This could be presented in a little table and would I think complement the qualitative data already presented.

Other than addressing these small points I would recommend acceptance.

Author Response

Point 1: This is a very good paper and I would be happy to recommend publication with one or two small adjustments.

Response 1: Thank you for these encouraging comments and suggestions, which we have followed.

Point 2: Firstly in the abstract lines 25-26 you should refer again to smaller urban green spaces being perceived as belonging to other people, rather than just green spaces.

Response 2: Done (lines 28-29):

The smaller green spaces were perceived as belonging to other people, which discouraged the older adults from using them.

Point 3: Methodologically you state a clear case for the combination of tools employed but there is no evidence of further approaches which you rejected. This would display greater critical thinking methodologically and should be included.

Response 3: Thank you for this suggestion. We now discuss why focus groups may be useful for further research on this topic (lines 569-575):

The use of focus groups may prompt discussions between participants on their use of green spaces, which could produce new insights into the reasons for differences in their experiences of larger and smaller green spaces. Additionally, this could prompt discussions on the history of the area, which may reveal how aspects of the socio-cultural context affect participants’ use of the green spaces. However, focus groups would have been less appropriate for our study since we were more interested in examining individual participants’ experiences of recent changes to specific green spaces.

Point 4: Lines 179-184 The interviewer questions seem to presuppose a positive response? Were participants permitted to say that they did not find spaces attractive? 'What do you like about this green space compared to other green spaces'? Did you also ask - Is there anything you do not like about this space? Some clarification here is required.

Response 4: Clarified to include questions from the topic guide which asked participants about aspects of the green space which they disliked or felt should be improved (lines 197-198):

In contrast, whilst visiting specific sites, the researcher asked questions such as “What is it about this site that makes it attractive to use?”, “Is there anything you dislike about this site?”, and “Are there any ways in which this space could be improved?”.

Point 5: How were the walk-round interviews recorded? Presumably using an audio recorder please clarify.

Response 5: Clarified (line 161):

For all participants, the interview was audio recorded using an encrypted Olympus audio recording device and transcribed verbatim.

Point 6: I understand that this was a qualitative study but I would be very interested to see a little quantification of the key themes and piling of the responses which fit within them. This could simply be a table which piles responses within sub-categories and then into the main themes. For instance within the theme of 'small urban spaces were not for them', what was the total amount of times that this was mentioned by participants? Clearly there were different reasons for this such as lack of amenity, lack of privacy etc. It would be interesting to see the frequency of these sub-themes within the main theme. This could be presented in a little table and would I think complement the qualitative data already presented.

Response 6: We have added a table (Table 2) showing the themes and the subthemes within each theme, with an example participant quote for each subtheme. In this table, we have included the number of participants who produced material contributing to each theme and subtheme (page 8, lines 280-281). However, we do not wish to quantify participants’ responses any further, since the aim of this research was to conduct a qualitative investigation of their experiences. Such research is not designed to estimate quantities of views, but to generate insights into these views.